# Role of Oxidative Stress in Peyronie’s Disease: Biochemical Evidence and Experiences of Treatment with Antioxidants

**DOI:** 10.3390/ijms232415969

**Published:** 2022-12-15

**Authors:** Gianni Paulis, Giovanni De Giorgio, Luca Paulis

**Affiliations:** 1Peyronie’s Care Center, Department of Urology and Andrology, Castelfidardo Clinical Analysis Center, 00185 Rome, Italy; 2Section of Ultrasound Diagnostics, Department of Urology and Andrology, Castelfidardo Clinical Analysis Center, 00185 Rome, Italy

**Keywords:** Peyronie’s disease, oxidative stress, antioxidants

## Abstract

Background: Peyronie’s disease (PD) is a chronic inflammatory condition affecting adult males, involving the tunica albuginea of the corpora cavernosa of the penis. PD is frequently associated with penile pain, erectile dysfunction, and a secondary anxious–depressive state. The etiology of PD has not yet been completely elucidated, but local injury is generally recognized to be a triggering factor. It has also been widely proven that oxidative stress is an essential, decisive component in all inflammatory processes, whether acute or chronic. Current conservative medical treatment comprises oral substances, penile injections, and physical therapy. Aim: This article intends to show how antioxidant therapy is able to interfere with the pathogenetic mechanisms of the disease. Method: This article consists of a synthetic narrative review of the current scientific literature on antioxidant therapy for this disease. Results: The good results of the antioxidant treatment described above also prove that the doses used were adequate and the concentrations of the substances employed did not exceed the threshold at which they might have interacted negatively with the mechanisms of the redox regulation of tissue. Conclusions: We believe new, randomized, controlled studies are needed to confirm the efficacy of treatment with antioxidants. However, we consider the experiences of antioxidant treatment which can already be found in the literature useful for the clinical practice of urologists in the treatment of this chronic inflammatory disease.

## 1. Introduction

Peyronie’s disease (PD) is a chronic inflammatory condition affecting adult males with an autosomal dominant genetic predisposition [1,2,3]. The prevalence of PD seems to vary by geographic area, and is between 11.0% and 13.0% in the USA, approximately 8.9% in Canada, 0.5% in Australia, 7.1% in Italy, 3.2% in Germany, 3.64% in Brazil, 0.6% in Japan, and 5.0% in China [4,5,6,7,8,9,10,11,12]. In an African epidemiological study, the prevalence of PD in black individuals was found to be between 0.1% and 3.5% [13].

The disease affects the tunica albuginea of the corpora cavernosa of the penis, and in its natural evolution can also involve the tissue of the corpora cavernosa and the intercavernous septum. The disease’s hallmark is an excessive production of collagen, which causes localized fibrosis and results in penile deformation. Most of the time, the clinical consequence is a penile bend that can be more or less pronounced, and is sometimes associated with penile shortening, penile torsion of various degrees, divots, hourglass deformity, etc. The disease is frequently associated with penile pain, erectile dysfunction, and a secondary anxious–depressive state. It has been found that PD patients are more likely to have psychiatric disorders (anxiety disorder, depression, substance abuse, alcohol abuse, self-injurious behavior, etc.) than men without PD [14,15]. It is therefore possible that psychological distress can negatively affect the correct daily intake of drugs and facilitate the progression of the disease. In PD patients, a depressive state is present in 48% of cases; it has also been found that psychological disorders can persist even after the successful treatment of PD [14,16].

The etiology of PD has yet to be completely elucidated, although the triggering factor is generally recognized to be local injury [17,18,19]. Initial injury (microtrauma or major trauma) is followed by the deposition of fibrin (small hematoma). Instead of being reabsorbed (as occurs in males who are not genetically susceptible to the disease), the hematoma leads to the recruitment of inflammatory cells and proinflammatory cytokines, which inevitably results in the formation of chronic inflammatory tissue that evolves towards fibrosis [20,21]. The presence of the following associated risk factors makes the onset of PD much more probable: erectile dysfunction, diabetes mellitus, high blood pressure, autoimmune disorders, chronic prostatitis, etc. [22].

It is possible that older PD patients may have other associated risk factors (hypertension, erectile dysfunction, diabetes, hyperlipidemia, etc.), which further favor the progression of the disease.

PD has two distinct stages: In the first (active) inflammatory stage, the inflammatory process causes the production of collagen, and the consequent remodeling of the tissue, resulting in fibrosis of the affected area (penile plaque). The plaque may then evolve towards calcification, which occurs in 20–25% of cases [23]. The first stage lasts for about 12–18 months [24,25]. The second stage consists of the stabilization of the disease. Inflammation is no longer present, pain is absent, and deformity has ceased to progress. Medical treatment is indicated in the disease’s first stage, while surgery is indicated in its second stage [25,26,27]. Current conservative medical treatment comprises the following: oral substances (vitamin E, colchicine, tamoxifen, potaba, phosphodiesterase-5 inhibitors, antioxidants (such as carnitine, propolis, etc.)); penile injections (corticosteroids, verapamil, pentoxifylline, clostridium histolyticum collagenase (CHC/Xiaflex), interferon, etc.); and physical therapy (extracorporeal shock wave therapy (ESWT), iontophoresis, penile traction devices and vacuum devices, etc.) [25,27,28,29,30,31,32,33,34,35,36]. 

This article, in addition to exposing the pathogenetic mechanisms of PD, consists of a synthetic narrative review of the current scientific literature on the antioxidant therapy of this disease. The aim of this article is to show how antioxidant therapy is able to interfere with the pathogenetic mechanisms of the disease.

## 2. Role of Oxidative Stress in Peyronie’s Disease

### 2.1. Pathophysiological and Biochemical Mechanisms of Peyronie’s Disease 

Oxidative stress has been widely proven to be an essential component in all inflammatory processes, whether acute or chronic [37,38,39,40,41]. Extensive studies have shown that oxidative stress plays a decisive role in Peyronie’s disease which, as we know, consists of chronic inflammation involving the tunica albuginea of the penile corpora cavernosa [20,21,42,43,44,45,46,47,48,49,50,51].

Following a traumatic event, which need not necessarily be violent, the tunica albuginea normally reacts with tissue repair and healing. However, when trauma occurs in a male who is genetically susceptible to PD, a series of events occurs which leads to the formation of fibrotic plaque. 

Delamination of the tunica albuginea as a result of the injury causes an effusion of blood, with a local accumulation of fibrin that is not reabsorbed due to insufficient fibrinolysis. Fibrin accumulation is a powerful factor in the local recruitment of inflammatory cells (macrophages, neutrophil granulocytes, T cells, mast cells, etc.), which immediately produce proinflammatory cytokines (interleukin-1 (IL-1), tumor necrosis factor-alpha (TNF-alpha)), and have chemotactic effects on fibroblasts [21]. 

Various cells that have been attracted to the site (fibroblasts, platelets, monocytes, macrophages, neutrophil granulocytes, and T cells) begin to produce important fibrogenic factors (transforming growth factor beta-1 (TGF-ß1), platelet-derived growth factor (PDGF), and basic fibroblast growth factor (bFGF)) which induce an overproduction of collagen in situ [21,52,53]. At the same time, macrophages and neutrophil granulocytes that are present in high concentrations at the inflammatory site, after being activated, commence to degranulate and produce large amounts of lysosomal enzymes and reactive oxygen species (ROS). 

### 2.2. Oxidative Stress in Peyronie’s Disease

The activation of these phagocytes occurs thanks to the enzyme nicotinamide adenine dinucleotide phosphate (NADPH oxidase) present in their cytoplasm [45,54,55]. The ROS released by the inflammatory cells are mainly superoxide anion (O_2_•-) and its metabolites of hydrogen peroxide (H_2_O_2_), hydroxyl radical (HO•), hypochlorous acid (HOCl) and singlet oxygen (^1^O_2_•) [54,55]. The local overproduction of ROS and proinflammatory cytokines induces the activation of nuclear factor kappa-B (NF-kB), a protein which regulates DNA transcription and, in particular, the gene expression of TGF-ß1, inducible nitric oxide synthase (iNOS), bFGF, fibrin, collagen, etc. [42,56]. 

A redundancy of biological signals is therefore established, producing, as a consequence, a local exacerbation of inflammation. The sources of production of the iNOS enzyme are macrophages, monocytes, T cells, smooth muscle cells, fibroblasts and myofibroblasts [43,45,57]. 

The production of iNOS is also stimulated by the cytokines IL-1 and TNF-alpha [58]. The iNOS enzyme is therefore locally produced in excess (100 to 1000 times more concentrated than normal constitutive NOS), leading to a chemical reaction and the production of high concentrations of nitric oxide radical (NO•-); i-NOS also bolsters the local synthesis of collagen [43,59]. This substance and its metabolites (peroxynitrite (ONOO-), peroxynitrous acid (HOONO), nitrogen dioxide radical (NO_2_•)) are also reactive species such as ROS and are called reactive nitroxidative species (RNS). 

These RNS, adding themselves to ROS, cause a further oxidation state, in this case nitro-oxidation, which can cause lipid peroxidation, DNA fragmentation, the nitration of proteins, and the alteration of vascular tone (vasoconstriction) [43,59]. All these biochemical events can cause greater tissue and cellular damage at the site of injury. It is therefore clear that a strong state of oxidation (oxidative and nitro-oxidative stress) created by the plentiful, prolonged local production of ROS and RNS, is a decisive factor for the progression and chronicization of inflammation [60,61]. The main biological mediators (and their properties) present in Peyronie’s disease are listed in Table 1: nuclear factor-B, i-NOS, TGF-ß1, PDGF, IL-1, bFGF, PAI-1, TNF-alpha, tissue inhibitors of metalloproteinases (TIMPs) [5,20,25,39,42,43,45,48,49,50,52,56,57,58,59,62,63,64,65,66,67,68,69,70,71,72,73,74,75,76,77,78,79,80,81,82,83,84,85,86,87,88] (see Figure 1).

## 3. Experiences of Treatment with Antioxidants in PD Patients

### 3.1. The First Studies on the Use of Antioxidants in Patients with PD

The use of antioxidants in the treatment of PD dates back over 70 years: specifically, vitamin E was the first antioxidant to be successfully used [89]. It was not chosen for its antioxidant properties, however, but merely because it had already been used with good results in other fibrotic disorders, such as Dupuytren’s contracture and primary fibrositis. At the time, it was postulated that fibrotic diseases, including PD, were due to a metabolic disorder caused by vitamin E deficiency [89]. 

Many years later, in recent years of the second millennium, propolis began to be used—likewise, not for its known antioxidant properties, but simply because a patient with PD, who had been treated with propolis (hospital in Havana, Cuba) by gastroenterologists because he was affected by Giardiasis, had noticed a significant improvement in his penile curvature. The urologists of the same hospital where the patient was treated, having observed the clinical evidence, began to treat PD patients with propolis (in oil form: propoleum). All the studies published by these Cuban urologists, which include both the exclusive use of propolis and treatment in association with laser therapy, showed significant improvement in terms of both plaque volume reduction and improvement in penile curvature [90,91,92,93]. 

Subsequently, in the early 2000s, carnitine was used to treat PD, specifically for its antioxidant properties, both alone (orally) and together with penile injections of a calcium channel blocker (Verapamil) [94,95]. Both studies achieved good results. 

A number of studies then followed in which pentoxifylline was successfully used to treat PD [29,96,97,98].

In other studies, a number of antioxidants (vitamin B3, propionyl-L-carnitine, L-arginine) were used in association with other substances (verapamil, tadalafil) or penile traction [46,99,100]. All of these studies achieved excellent results. Furthermore, these studies introduced the concept of “combined” or “multimodal” antioxidant treatment to the literature.

### 3.2. Multimodal Treatment with Antioxidants 

Multimodal or combined treatment is a therapeutic practice which has already been used in other fields of medicine, such as oncology (polychemotherapy) and the treatment of infections (antibiotic combination therapy), for instance, in the treatment of tuberculosis or even simply in the treatment of drug-resistant bacteria, e.g., for amoxicillin-resistant germs, through the association of amoxicillin with clavulanic acid. The aim is to obtain a better therapeutic result than is achievable by administering a single substance. 

In our case, this is particularly advantageous, since many antioxidants also have anti-inflammatory and antifibrotic properties that differ slightly (see Table 2A,B), and therefore combining them makes it possible to reduce the dose of each antioxidant (and minimize the possibility of adverse effects from an overdose) and to tackle PD and its various biochemical mechanisms, interfering in different ways with the many “chemical messengers” involved. 

For instance, bilberry is capable of inhibiting the production of bFGF, unlike other antioxidant substances. Boswellia, superoxide dismutase and diclofenac cannot reduce the synthesis of PAI-1 as other substances do. Pentoxifylline, silymarin, boswellia, coenzyme Q10, carnitine, and Ginkgo biloba, unlike other antioxidant substances, are able to contrast the vasoconstriction caused by high concentrations of nitroxide radicals (NO•-), and thanks to their properties, they can cause vasodilation at the disease site [101,102], thus preventing hypoxia and tissue damage and contrasting the outcomes of PD.

Coenzyme Q10 is capable of activating NF-E2-related factor-2 (Nrf2) which suppresses TGF-β1 (fibrogenic factor) expression [103,104]. 

Thus, it appears that multimodal antioxidant therapy can be used as a therapeutic strategy to maximize the end effects of treatment. 

It must be specified that propolis is a product made by bees containing many substances that have useful properties for PD treatment: polyphenols (including flavonoids and resveratrol), cinnamic acid, caffeic acid, fatty acids, certain vitamins (E, C, B1, B2, and B6), and chemical elements (zinc, iron, calcium, sodium, potassium, magnesium, manganese, copper, iodine, nickel, titanium, cobalt, and silicon). The most important flavonoids contained in propolis are the following: galangin, quercetin, apigenin, acacetin, catechin, chrysin, luteolin, kaempferol, pinocembrin, myricetin, naringenin, and rutin [105]. The geographical area and climate from which propolis comes can influence its composition, since the plants from which bees extract resins (which they then process) can be different [105]. To get around this problem, the pharmaceutical industry, in preparing propolis as a product, selects raw materials based on their characteristic properties. 

Table 2A,B list the antioxidants we use in our multimodal treatment: vitamin E, vitamin C, propolis, bilberry, silymarin, Ginkgo biloba, carnitine, coenzyme Q10, boswellia, pentoxifylline, superoxide dismutase, hyaluronic acid, diclofenac. 

Table 2A,B also report the biochemical properties and mechanisms of action of the listed antioxidant substances [101,102,106,107,108,109,110,111,112,113,114,115,116,117,118,119,120,121,122,123,124,125,126,127,128,129,130,131,132,133,134,135,136,137,138,139,140,141,142,143,144,145,146,147,148,149,150,151,152,153,154,155,156,157,158,159,160,161,162,163,164,165,166,167].

**Table 2 ijms-23-15969-t002:** (**A**). Biochemical properties and molecular mechanisms of the substances used in the multimodal antioxidant therapy of Peyronie’s disease. Panel for effective understanding of “targeted antioxidant therapy” (Part 1). (**B**). Biochemical properties and molecular mechanisms of the substances used in the multimodal antioxidant therapy of Peyronie’s disease. Panel for effective understanding of “targeted antioxidant therapy” (Part 2).

**(A)**			
**Substance**	**Biochemical Property**	**Molecular Mechanism**	**References**
Vitamin E	Antioxidant Anti-inflammatory Antifibrotic Antiplatelet aggregation	It inhibits the production of reactive oxygen species (by inflammatory cells). It inhibits the production of reactive nitroxidative species. It inhibits the activation of NF-kappa-B. It reduces pro-inflammatory cytokine and PAI-1 production. It inhibits COX-2 activity. It inhibits proliferation of human fibroblasts. It inhibits platelet aggregation.	[106,107,108,109,110,111]
Vitamin C	Antioxidant Anti-inflammatory Antifibrotic Regenerative action on Vitamin E	It scavenges reactive oxygen species and reactive nitroxidative species. It is a potent scavenger of superoxide anion, hydroxyl radical, singlet oxygen, and lipid hydroperoxides. It reduces pro-inflammatory cytokine and PAI-1 production. It inhibits fibrosis (deposition of collagen) by inhibiting TGF-β1. It regenerates vitamin E in its normal and non-oxidized form.	[112,113,114,115]
Propolis	Antioxidant Anti-inflammatory Antifibrotic Antiplatelet aggregation	It scavenges reactive oxygen species and reactive nitroxidative species. It hinders neutrophil migration to the site. It scavenges superoxide anion, hydroxyl radical, singlet oxygen, hypochlorous, lipid hydroperoxides and peroxynitrite. It inhibits COX-1 and COX-2 activity. It reduces production of: inducible-NOS, NF-kappa-B, and some cytokines (IL-1, IL-6, IL-8, TNF-α, TGF-β1). It reduces production of PDGF, PAI-1 and fibronectin. It inhibits fibrosis (deposition of collagen) by inhibiting TGF-β1 and PDGF. It inhibits myofibroblastic differentiation of fibroblasts. It inhibits MMP-2 and MMP-9 (with elastase activity). It inhibits platelet aggregation.	[105,116,117,118,119,120,121,122,123,124,125]
Bilberry	Antioxidant Anti-inflammatory Antifibrotic	It scavenges reactive oxygen species and reactive nitroxidative species. It scavenges superoxide anion, hydroxyl radical, singlet oxygen, hypochlorous, lipid peroxyl radicals, and peroxynitrite. It inhibits COX-2 activity. It reduces production of: i-NOS, NF-kappa-B, and some cytokines (IL-1, IL-6, IL-8, TNF-α, TGF-β1). It reduces production of PDGF, bFGF and PAI-1. It inhibits fibrosis (deposition of collagen) by inhibiting TGF-β1, PDGF, and bFGF. It inhibits MMP-2 and MMP-9 (with elastase activity).	[126,127,128]
Silymarin	Antioxidant Anti-inflammatory Antifibrotic Vasorelaxant Antiplatelet aggregation	It scavenges reactive oxygen species and reactive nitroxidative species. It scavenges superoxide anion, hydroxyl radical, singlet oxygen, hypochlorous, lipid peroxyl radicals and peroxynitrite. It inhibits COX-2 activity. It reduces production of: i-NOS, NF-kappa-B, some cytokines (IL-1, IL-6, IL-8, TNF-α, TGF-β1), and PAI-1. It inhibits fibrosis (deposition of collagen) by inhibiting TGF-β1. It inhibits platelet aggregation. It determines vasodilation through an endothelial mechanism via the nitric oxide pathway.	[129,130,131,132]
Ginkgo biloba	Antioxidant Anti-inflammatory Antifibrotic Vasorelaxant Antiplatelet aggregation	It scavenges reactive oxygen species and reactive nitroxidative species. It reduces production of: i-NOS, NF-kappa-B, IL-1, IL-6, IL-10, TNF-α, TGF-β1, and PAI-1 It hinders inflammation, inhibiting: NF-kappa-B and pro-inflammatory cytokine production, and COX-2 activity It inhibits fibrosis (deposition of collagen) by inhibiting TGF-β1 It determines vasodilation through an endothelial mechanism, via the nitric oxide pathway. It inhibits platelet aggregation	[133,134,135,136]
**(B)**			
**Substance**	**Biochemical Property**	**Molecular Mechanism**	**References**
Carnitine	Antioxidant Anti-inflammatory Antifibrotic Vasorelaxant Antiplatelet aggregation	It scavenges reactive oxygen species and reactive nitroxidative species. It reduces production of: i-NOS, NF-kappa-B, IL-1, IL-6, IL-8, TNF-α, TGF-β1 and PAI-1. It hinders inflammation, inhibiting: NF-kappa-B and pro-inflammatory cytokine production, and COX-2 activity. It inhibits fibrosis (deposition of collagen) by inhibiting TGF-β1. It inhibits fibroblast proliferation and the osteoblastic differentiation of fibroblasts. It inhibits myofibroblastic differentiation of fibroblasts by inhibiting TGF-β1. It determines vasodilation through an endothelial mechanism, via the nitric oxide pathway. It inhibits collagen-induced platelet aggregation.	[137,138,139,140,141,142]
CoenzymeQ10	Antioxidant Anti-inflammatory Antifibrotic Vasorelaxant Antiplatelet aggregation Regenerative action on Vitamin E and Vitamin C	It scavenges reactive oxygen species and reactive nitroxidative species. It protects cell membranes from lipoperoxidation determined by reactive species. It reduces production of: i-NOS, NF-kappa-B, IL-1, IL-6, IL-10, TNF-α, TGF-β1 and PAI-1. It inhibits fibrosis (deposition of collagen) by inhibiting TGF-β1. It activates NF-E2-related factor-2 (Nrf2) which suppresses the TGF-β1 expression. It inhibits MMP-2 and MMP-9 (with elastase activity). It inhibits platelet aggregation. It determines vasodilation through an endothelial mechanism via the nitric oxide pathway. It regenerates Vitamin E in its normal and nonoxidized form.	[143,144,145,146,147,148]
Boswellia	Antioxidant Anti-inflammatory Antifibrotic Vasorelaxant Antiplatelet aggregation	It scavenges reactive oxygen species and reactive nitroxidative species. It protects cell membranes from lipoperoxidation determined by reactive species. It reduces production of: i-NOS, IL-1, IL-6, TNF-α, and TGF-β1. It hinders inflammation, inhibiting: NF-kappa-B and pro-inflammatory cytokine production, and COX-2 activity. It inhibits fibrosis (deposition of collagen) by inhibiting TGF-β1. It determines vasodilation through an endothelial mechanism via the nitric oxide pathway. It inhibits platelet aggregation.	[149,150,151]
Pentoxifylline	Antioxidant Anti-inflammatory Antifibrotic Vasorelaxant Antiplatelet aggregation	It scavenges reactive oxygen species and reactive nitroxidative species. It reduces production of: TNF-α, i-NOS, IL-1, IL-6, IL-8, IL-10, TGF-ß1, PDGF, and PAI-1. It hinders inflammation, inhibiting: NF-kappa-B and pro-inflammatory cytokine production. It inhibits fibrosis (deposition of collagen) by inhibiting TGF-β1 and PDGF. It determines vasodilation through a nonselective PDE inhibition (preventing the degradation reaction of cyclic AMP). It inhibits platelet aggregation.	[101,102,152,153,154,155,156,157]
Superoxide dismutase	Antioxidant Anti-inflammatory Antifibrotic	It defends the human body against tissue damage mediated by reactive oxygen species (ROS). It eliminates superoxide anion (O2•-). It inhibits neutrophil-induced inflammation. It hinders fibrosis (collagen deposition) by downregulating TGF-ß1.	[158,159]
Hyaluronic acid	Antioxidant Anti-inflammatory Antifibrotic Antiplatelet aggregation	It scavenges ROS, superoxide anion, and hydroxyl radicals. It inhibits lipid peroxidation. It hinders inflammation reducing production of: TNF-α, IL-6, IL-1, and PAI-1. It inhibits fibroblast proliferation. It inhibits platelet aggregation.	[160,161,162,163,164]
Diclofenac	Anti-inflammatory Antioxidant	Like other nonsteroidal anti-inflammatory drugs (NSAIDs), diclofenac inhibits synthesis of prostaglandins by inhibiting COX-1 and COX-2 activity. It inhibits NF-kappa-B gene expression. It inhibits TNF-induced NF-kappa-B activation. It exerts powerful dose-dependent free-radical-scavenging activity. It strongly protects against lipid peroxidation and the damage of peroxyl radicals.	[165,166,167]

### 3.3. Brief Narrative Review of Peyronie’s Disease Treatment with Antioxidants

Although a number of guidelines contain no recommendation for the use of antioxidants in the treatment of PD, they have been used in several therapeutic experiences, either alone or in combination with other substances [168,169].

We searched the PubMed database for articles on the topic “antioxidant treatment in Peyronie’s disease” and found 21 articles on this topic. We considered the following articles eligible: randomized and/or controlled clinical trials; case reports containing more than one case report. After screening the 21 articles, we excluded 4 of them: 2 articles because they reported a single clinical case, and 2 clinical studies because they did not have a control group.

We shall, of course, cite only a selection of the scientific literature on the topic, including randomized studies and a number of controlled studies, as well as three case report studies in which the complete regression of plaque was achieved [29,30,98,170,171,172,173,174,175,176]. Other minor studies cited in the References section of this article are not described here for reasons of space, despite the fact that most of them are controlled trials and despite their positive results [90,91,92,93,94,95,96,97,177,178,179] (Figure 2).

A randomized, double-blind, placebo-controlled study by Riedl (2005) used *Superoxide dismutase* to treat PD patients. Superoxide dismutase was used as topical gel once a day for eight weeks, reaching the following statistically significant results: pain reduction in 52.6% of treated cases, compared to 20% in the control group; plaque volume reduction in 47% of treated cases, compared to the control group where, on the contrary, plaque volume grew in 8% of cases; reduction in the degree of penile curvature (between 5 and 30 degrees) in 23% of treated cases, compared to the control group in which, instead, curvature increased in 10% of cases [170].

In a randomized controlled study by Favilla et al. (2014), the authors treated PD patients for 12 weeks with oral antioxidants (vitamin E, para-aminobenzoic acid, propolis, blueberry anthocyanins, soy isoflavones, muira puama, damiana, persea americana) associated with weekly verapamil injections. The control group only received weekly verapamil injections [171]. After treatment, there were no statistically significant differences for some endpoints (e.g., reduction in plaque size = −29% (group receiving oral antioxidants + verapamil injections) and −38% (group receiving only verapamil injections); improvement in penile curvature = −11.9 degrees (group receiving oral antioxidants + verapamil injections) and −10.8 degrees (group receiving only verapamil injections)).

Regarding the other endpoints, in the group receiving oral antioxidants + verapamil injections, there were statistically significantly better results compared to the control group (only verapamil injections) in terms of: penile pain reduction, orgasmic function, intercourse satisfaction, and overall satisfaction. In conclusion, the study showed that combined therapy made it possible to obtain better results [171].

In a randomized controlled study by Alizadeh et al. (2014), the authors treated patients with PD for six months, dividing them into three treatment groups:First group: only oral therapy with pentoxifylline;Second group: only intralesional injections with verapamil (every other week, 12 total injections);Third group: oral pentoxifylline + intralesional injections with verapamil (every other week, 12 total injections) [98].

The results were as follows: improvement in penile curvature: first group = 26.7%, second group = 36.7%, third group = 36.7%; plaque size reduction: first group = 30%, second group = 33%, third group = 33%; improvement in erectile dysfunction: first group = 46.7%, second group = 66.7%, third group = 86.7%; penile pain reduction: first group = 73.3%, second group = 76.7%, third group = 80%. Even in this study, the results show that combined therapy enables better results [98].

In their retrospective control group study, Gallo et al. (2019) treated PD patients for 6 months, dividing them into three groups: first group: oral therapy with arginine and pentoxifylline; second group: oral therapy with arginine and pentoxifylline + intralesional injections with verapamil (every other week, 12 total injections); third group: oral therapy with arginine and pentoxifylline + intralesional injections with verapamil (every other week, 12 total injections) + penile traction therapy with a penile extender, applied daily for about 2–8 h [172]. The therapeutic results for the various endpoints, compared with the baseline data, were as follows:–Reduction in curvature by at least 10 degrees: first group = 0%; second group = 17.8%; third group = 50%.–Improvement in the International Index of Erectile Function (IIEF) score (normal score > 25): first group = from 17.7 to 18.5 (increase + 0.8); second group = from 20.4 to 21.6 (increase + 1.2); third group = from 20 to 22.4 (increase + 2.2).–Change in stretched penile length: first group = from 10.5 to 10.4 cm (reduction −0.1 cm); second group = from 10.7 to 10.6 cm (reduction −0.1 cm); third group = from 10.3 to 11.0 cm (increase + 0.7 cm).–Penile pain: first group = resolution of pain in 100% of cases; second group = resolution of pain in 100% of cases; third group = resolution of pain in 100% of cases [172].

The authors conclude that oral therapy alone can simply block disease progression, association between oral therapy and verapamil injections enables only slight improvement, and the combination of oral therapy, verapamil injections, and penile traction therapy is the only conservative approach leading to optimal results.

In our 2016 retrospective control group study (Paulis et al., 2016), we treated 206 patients with PD using the following combined therapy: group A: 112 patients, pentoxifylline (perilesional injections) twice a month for 6 months + oral pentoxifylline + oral propolis + oral blueberry + oral vitamin E + diclofenac gel for 6 months); group B: 94 patients undertaking the same therapy as group A but with no pentoxifylline injections; group C: 101 patients with PD and no treatment [30].

After treatment, a better response was observed in group A, where the combined therapy was bolstered by the association with pentoxifylline injections: a reduction in plaque volume in 100% of cases compared to the 79.7% obtained in group B; a mean reduction in plaque volume in 46.9% of cases compared to 24.8% of group B; an improvement in curvature in 96.8% of cases compared to 56.4% of group B; a mean reduction in the angle of the curve = −10.1 degrees compared to −4.8 degrees in group B; recovery of normal penile rigidity in patients with ED was obtained in 56% of cases compared to 23.5% of group B [30].

In 2013, we published a controlled study (Paulis et al., 2013) which, in contrast with other articles in the literature, proved that oral vitamin E, associated with other substances (oral propolis + oral blueberry + verapamil injections) + topical diclofenac (multimodal therapy) for 6 months, was statistically effective in curing PD. In particular, vitamin E was shown to improve the therapeutic result: increasing the reduction in the volume of plaque after treatment (from 35.8% to 50.2%); increasing the percentage of patients who achieved an improvement in their penile curvature after treatment (from 48.4% to 96.6%); and increasing the degree of reduction in the curve (from 6.7 degrees to 12.2 degrees) [173].

Finally, we cite three case report articles of ours in which, overall, thanks to multimodal antioxidant treatment, we obtained the complete regression of plaque in eight patients suffering from PD [174,175,176].

In all patients, we used most of the antioxidants mentioned above (vitamin E, vitamin C, propolis, bilberry, silymarin, Ginkgo biloba, coenzyme Q10, boswellia, superoxide dismutase, diclofenac and pentoxifylline). In two cases, the combined oral and topical therapy was not associated with pentoxifylline injection therapy because the patients did not give their consent.

Treatment duration varied from 28 months to 53 months, most likely depending on plaque volume, since the patient who was cured in 28 months initially had plaque which was measured by ultrasound to be 122 mm^3^, while the patient whose treatment took 53 months had an initial plaque volume measuring 733 mm^3^ and received no penile injections because of his refusal to undergo injection therapy.

The time period required for treatment was necessarily long due to the nature of the disease, i.e., its being a chronic (not an acute) inflammation, which therefore requires adequate treatment time. All patients, in any case, were informed before starting treatment that treatment would have to be long, due to the intrinsic characteristics of PD.

In all cases, at the end of treatment, patients experienced the complete regression of both their baseline penile pain and the penile curvature caused by the disease.

## 4. Discussion

Oxidative stress evidently plays a leading role in Peyronie’s disease, a disease where there is an evident redox imbalance caused by the production of reactive species, following trauma to the penis in a genetically predisposed male.

It is also evident that oxidative stress plays a very large role not only in the onset, but especially in the progression of PD, resulting in local tissue damage and the formation of fibrotic plaque and its possible calcification [20,21,23].

From the literature, it is therefore clear that the most important physiopathogenic events are the following: penile injury (including minor trauma), formation of blood collection in the corpus cavernosum, recruitment of inflammatory cells, release of ROS and cytokines, activation of transcription factor NF-kB, the production of iNOS and the subsequent overproduction of radical NO, and the overproduction and local deposition of collagen (plaque formation) (Figure 1). All the antioxidants we use in the treatment of PD, as indicated in Table 2A,B, can interfere with the disease’s basic pathophysiological mechanisms, and this explains the good therapeutic outcomes in the literature.

The good results of the antioxidant treatment described above also prove that the doses used were adequate and the concentrations of the substances employed did not exceed the threshold beyond which they might have instead interacted negatively with the mechanisms of redox regulation of the tissues (see “antioxidant paradox”) [180,181]. It is very likely that larger doses would have rendered the treatment ineffective or—worse still—favored disease progression.

Several antioxidant substances, some of which we use to treat PD, have been successfully used in other PD-related diseases. Silymarin, when given in a dose of 140 mg thrice daily for 3 months as an adjuvant for glycemic control, lipid profile, and insulin resistance, proved to have a beneficial efficacy [182]. Additionally, Daflon 500 mg (micronized a purified flavonoid fraction of Rutaceae aurantiae, consisting of 90% diosmin and 10% hesperidin) given twice daily for 45 days is helpful in reducing glucose level and the risk of cardiovascular disease [183].

In our clinical practice, in particular in the treatment of PD, we never doubted that our antioxidant doses were adequate and not excessive, especially since in our first experiences, we used at most three or four substances (vitamin E, propolis, blueberry, diclofenac) [173,177]. Over the following years, we gradually introduced other antioxidant substances in our combined PD therapy, with the aim of obtaining even more positive results; we always used low doses, even for these additional substances. Since in the past few years the results of our treatments have proved ever more effective and with no appreciable side effects, we are convinced we are not causing any damage to the redox system. On the contrary, the excellent results achieved in our most recent studies confirm that the most effective treatment for PD is a multimodal treatment, in which the various substances combine, with their different individual properties, in contrasting the multiple physiopathogenic mechanisms at play, with the aim of achieving better therapeutic results than those which could be obtained using a single or a very small number of substances [184].

We believe that many current articles on the conservative treatment of PD are lacking, and unable to accurately evaluate the volume of plaque before and after the treatment. The plaque volume must be evaluated only with a very sensitive ultrasound system and a three-dimensional evaluation with the formula of the ellipsoid (volume = 0.524 × length × width × thickness). The change in plaque volume is an important endpoint that should always be evaluated, together with the other parameters, for a correct evaluation of the therapeutic result. We propose that all future studies concerning the conservative treatment of PD include this very important parameter.

## 5. Conclusions

Oxidative stress represents the fundamental chemical environment for the onset and progression of Peyronie’s disease, and—more broadly—for chronic inflammatory states, degenerative diseases, and malignant neoplasms. We think the good clinical results obtained using antioxidant therapy confirm the important role played by oxidative stress in PD. We also believe that the experiences of antioxidant treatments already present in the literature represent an important aid to the clinical practice of urologists for the treatment of this chronic inflammatory disease which, incidentally, is not rare, as its prevalence, especially in the Western world, is similar to that of diabetes mellitus (=10.5% worldwide) [185].

Treating PD with antioxidants directly interferes with the most important mechanisms of inflammation, thus treating the disease directly and not just its symptoms. We therefore hold that PD is a disease which can be cured, in contrast to the affirmation of authors who consider it an incurable condition [186]. We advocate this position in view of the fact that in the field of andrology, the prevailing background belief is that the gold standard of PD is surgery—basically, “treating the curve” and not “treating the disease”—whereas we and other authors support the latter [187].

We recommend that only patients with active (unstabilized) PD be treated with medical therapy. In the case of large penile plaques, we recommend adding periodic perilesional injections with pentoxifylline to the oral antioxidant substances.

In any case, we believe further studies are needed to both improve the understanding of the pathophysiology of Peyronie’s disease and to implement new randomized controlled trials to confirm the efficacy of treatment with antioxidants. We also propose multi-center studies involving the combined use of antioxidants in order to demonstrate more clearly that PD can be cured.

## Figures and Tables

**Figure 1 ijms-23-15969-f001:**
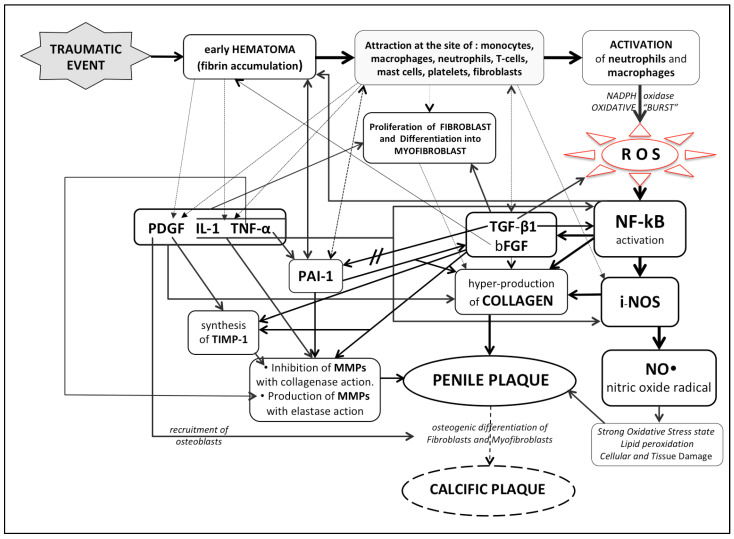
Explanatory algorithm of the pathophysiology of Peyronie’s disease.

**Figure 2 ijms-23-15969-f002:**
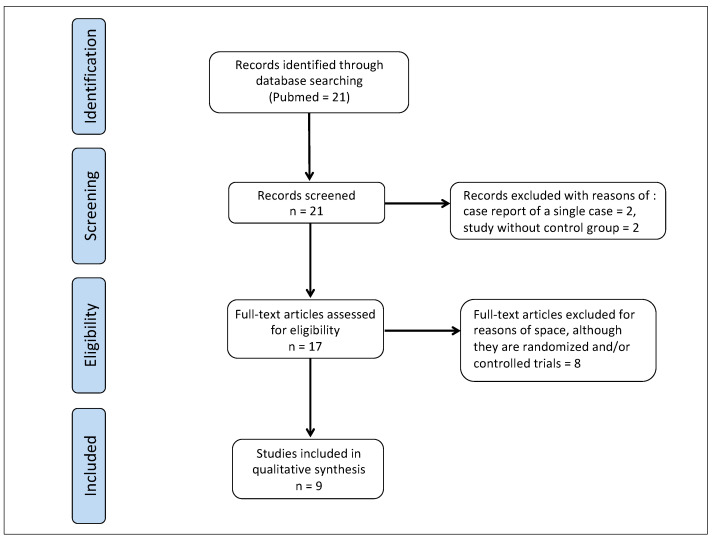
Explanatory flow chart of the narrative revision methodology.

**Table 1 ijms-23-15969-t001:** List of major biological mediators (and their properties) involved in Peyronie’s disease. Useful table for the choice of a possible “targeted antioxidant therapy”.

Name	Acronym	Activity	Cellular Source	Positive Regulators
Nuclear factor-B	NF-kB	It controls the transcription of DNA. It induces gene overexpression of TGF-ß1, iNOS, bFGF, fibrin, collagen, etc.	All cell types	ROS, TGF-beta-1, IL-1, TNF-alpha
Inducible nitric oxide synthase	i-NOS	It induces local overproduction of nitric oxide radical (NO •). It enhances collagen synthesis.	Macrophages, monocytes, T-lymphocytes, smooth muscle cells, fibroblasts, myofibroblasts	NF-kB, IL-1, TNF-alpha
Transforming growth factor beta-1	TGF-beta-1	Chemotactic action on neutrophils, monocytes, lymphocytes, and fibroblasts. It induces the production of collagen by fibroblasts. It stimulates the proliferation of fibroblasts and the transformation of fibroblasts into myofibroblasts. It induces collagen synthesis and deposition. It induces the production of ROS. It increases the synthesis of tissue inhibitors of matrix metalloproteinase (TIMP-1). It inhibits the production of matrix metalloproteinases (MMPs) with collagenase action (MMP-1, MMP-8, and MMP-13). It induces the production of MMP-2 and MMP-9 (with elastase activity). It induces the activation of NF-kB. It induces osteogenesis in PD plaque. It inhibits production of plasminogen activator inhibitor-1 (PAI-1).	Platelets, macrophages, neutrophils, T-lymphocytes	NF-kB, Reactive oxygen species (ROS), PAI-1
Platelet-derived growth factor	PDGF	Chemotactic action on fibroblasts. It induces the production of TIMP-1 and MMP-2 (with elastase activity). It induces collagen synthesis and deposition. It stimulates the proliferation of fibroblasts and the transformation of fibroblasts into myofibroblasts. It contributes to plaque calcification and ossification. Furthermore, it acts as an osteoblast recruiter.	Platelets and macrophages	Local accumulation of fibrin
Interleukin-1	IL-1	Chemotactic action on fibroblasts. It induces collagen synthesis and deposition. It stimulates bFGF and iNOS production. It induces the activation of NF-kB. It increases the production of MMPs.	Macrophages and fibroblasts	Thrombin in the damaged site
Basic fibroblast growth factor	bFGF	Chemotactic action fibroblasts. It stimulates the proliferation of fibroblasts. It induces collagen synthesis and deposition. It increases the synthesis of tissue inhibitors of matrix metalloproteinase (TIMP-1). It induces the production of MMP-2, MMP-9 (with elastase activity). It determines the further deposition of fibrin on site.	Fibroblasts, myofibroblasts, T-lymphocytes	Nuclear factor kappa-B (NF-kB), IL-1 PAI-1
Plasminogen activator inhibitor-1	PAI-1	It inhibits fibrinolysis by determining the persistence of fibrin in loco and triggering the recruitment of inflammatory cells. It stimulates the release of profibrogenic factors (cytokines, etc.) and then it induces collagen synthesis and deposition. It hinders collagenolysis by inhibiting MMPs with collagenase action. It increases the synthesis of MMP-9 (with elastase activity).	Platelets, endothelial cells, smooth muscle cells, fibroblasts, monocytes, macrophages	Thrombin in the damaged site TNF-alpha
Tumor necrosis factor-alpha	TNF-alpha	It induces the synthesis of PAI-1. It stimulates the proliferation of fibroblasts. At high concentrations it stimulates collagenase synthesis in fibroblasts. It stimulates iNOS production. It induces the activation of NF-kB. It increases the synthesis of MMP-9 (with elastase activity). It induces cellular apoptosis.	Monocytes, macrophages, T-lymphocytes	Fibrinogen and fibrin
Tissue inhibitors of metalloproteinases	TIMPs	They inhibit matrix metalloproteinases (MMPs). They regulate the connective tissue metabolism.	Many cell types (monocytes, macrophages, vascular smooth muscle cells, fibroblasts)	TGF-beta-1, PDGF, bFGF
Matrix metalloproteinases (MMPs)	MMP-1, MMP-2, MMP-8, MMP-9, MMP-10, MMP-12, MMP-13, MMP-18	MMP-2, MMP-9, MMP-10, MMP-12 (with elastase activity). MMP-1, MM- 8, MMP-13, MMP-18 (with collagenase action). MMPs can regulate cytokine activity.	Fibroblasts, myofibroblasts, neutrophils, macrophages, endothelial cells, vascular smooth muscle cells	IL-1, bFGF, TNF-alpha, PAI-1, PDGF, TGF-beta-1

## Data Availability

Not applicable.

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
