# Peer review of "Role of Oxidative Stress in Peyronie’s Disease: Biochemical Evidence and Experiences of Treatment with Antioxidants"

_ijms, 2022, doi:10.3390/ijms232415969_

Round 1

Reviewer 1 Report

This is an informative review that highlights the key role of oxidative stress in Peyronie’s Disease and lists different human studies where PD was treated with antioxidants. The authors’ advocacy of the view that PD can be cured is also very helpful to the field. I suggest the following changes to enhance the quality of the manuscript:

1.       In Introduction, the authors describe that in males who are genetically susceptible to the disease, “the hematoma leads to recruitment of inflammatory cells and pro-inflammatory cytokines, which inevitably results in the formation of chronic inflammatory tissue that evolves towards fibrosis”. Do local injuries in other tissues or organs lead to similar progression in patients with this genetic predisposition, or this phenomenon is unique to penis? 

2.       Have any studies characterized the function of antioxidant signaling pathways, such as Nrf2, in these tissues during disease progression? If so, I strongly recommend adding this information to the manuscript. 

3.       To make this article more practical and helpful, can the authors provide some specific recommendations or treatment guidelines for clinicians based on the aggregate results of the human studies reviewed in this manuscript?

4.       In Table 1, “Source of activation” can be misleading. I suggest changing it to “Positive Regulator(s)”.

5.       Fig.1 needs more legend to explain the diagram. For example, what do different types of lines, such as solid vs. various dotted lines, indicate? 

6.       It would be very helpful if the authors summarize the human studies in a table listing parameters for each study including study population, length of study, interventions used in different arms, end points, results, etc. The audience can much more readily and clearly visualize and compare all the studies if the information is presented in a table.

7.       Can the authors identify some overall takeaways and gaps from the human studies conducted so far? Can the authors propose designs for new trials that can fill these gaps and may promote the view that PD is curable through antioxidant therapies?  

Author Response

Reply to Reviewer #1

IMPORTANT:  Manuscript with corrections also of English by the MDPI linguistic service 

  1. In Introduction, the authors describe that in males who are genetically susceptible to the disease, “the hematoma leads to recruitment of inflammatory cells and pro-inflammatory cytokines, which inevitably results in the formation of chronic inflammatory tissue that evolves towards fibrosis”. Do local injuries in other tissues or organs lead to similar progression in patients with this genetic predisposition, or this phenomenon is unique to penis?

ANSWER

Thanks for your request for clarification. This phenomenon only affects patients with Peyronie's disease who always have a genetic predisposition.

  1. Have any studies characterized the function of antioxidant signaling pathways, such as Nrf2, in these tissues during disease progression? If so, I strongly recommend adding this information to the manuscript.

ANSWER

Thank you for your request. There are two articles in the literature on this topic.

Coenzyme Q10 activates NF-E2-related factor-2 (Nrf2) which suppresses the TGF-β1 expression. I added this information in the manuscript and in the table.

  1. To make this article more practical and helpful, can the authors provide some specific recommendations or treatment guidelines for clinicians based on the aggregate results of the human studies reviewed in this manuscript?

ANSWER

Thank you for your request. I have included a short text on the subject in the Conclusions section.

  1. In Table 1, “Source of activation” can be misleading. I suggest changing it to “Positive Regulator(s)”.

ANSWER

Thanks for the tip. I changed the name and entered “Positive Regulators” in Table 1.

  1. Fig.1 needs more legend to explain the diagram. For example, what do different types of lines, such as solid vs. various dotted lines, indicate?

ANSWER

Thank you for your request. I changed the legend to better explain the diagram. The different colors in the tables indicate nothing, they were only for division of the tabular sections. However, I have removed the colors.

  1. It would be very helpful if the authors summarize the human studies in a table listing parameters for each study including study population, length of study, interventions used in different arms, end points, results, etc. The audience can much more readily and clearly visualize and compare all the studies if the information is presented in a table.

ANSWER

Thank you for your request. I have included in the manuscript a table where one can easily understand the methodology of this narrative review.

I also did this because it was requested by the other Reviewer, in order to enrich the article.

  1. Can the authors identify some overall takeaways and gaps from the human studies conducted so far? Can the authors propose designs for new trials that can fill these gaps and may promote the view that PD is curable through antioxidant therapies?

ANSWER

Thank you for your suggestion. On the subject I have inserted new text in the manuscript, in the Discussion and Conclusions sections.

------------------------------------------------------------------------------

Reviewer 2 Report

The manuscript is very interesting dealing with disease of high significance on men health and the information of the antioxidants and anti-inflammatory substances are of highly benefit not only to Peyronie’s disease (PD) but also to many other diseases.

The manuscript was well written and require revision concerning the below comments and suggestions to be performed.

1-The abstract better to be divided into background, aim, method, results, and conclusion.

2-The following sections were absent; methods and Results although they are present as information but not under methods and results.

3-Captions of tables and figures required to be written in full details explaining the target from the table or figure, the importance and effect of information on the PD.

4- Effect of age and psychiatric disorders on the progression of PD better to be mentioned in the review.

5-Study flow chart for the review is missed which would be much better to be designed and submitted with the review as a summarized explanatory chart for the methodology.

6-The insulin resistance and glycemic control are important etiology of many inflammatory diseases. Hereunder paragraphs containing  some antioxidants with their mechanisms  as silymarin, diosmin,hesperidin, coenzyme Q10 and alpha-tocopherol, and genistein. So, the following paragraph to be added to the introduction of the manuscript ;

"Suboptimal glycemic control and insulin resistance are associated with high risk of
macrovascular and microvascular complications. Silymarin is a herbal medicine with an antioxidant and anti-inflammatory properties when given in a dose of 140mg thrice daily for 3 months as an adjuvant for glycemic control, lipid profile and insulin resistance proved to have a beneficial efficacy. [1] Additionally, Daflon 500 mg (micronized purified flavonoid fraction of Rutaceae aurantiae, consisting of 90% diosmin and 10% hesperidin), twice daily for 45 days is helpful in reducing glucose level and the risk of cardiovascular disease. [2]

Additionally,  herbs, and healthy diet including fruits and vegetables, can supply the body with beneficial nutrients and antioxidants [3] including coenzyme Q10 and alpha-tocopherol which have a neuroprotective as they proved to have protective effects of antioxidants [4]. Moreover, estrogenic compounds as genistein proved to exhibit a neuroprotective effect attributed to its estrogenic, antioxidant, and/or anti-apoptotic properties [5].

References

[1] Amany Talaat Elgarf, Maram Maher Mahdy, Nagwa Ali Sabri. Effect of Silymarin Supplementation on Glycemic Control, Lipid Profile and Insulin Resistance in Patients with Type 2 Diabetes Mellitus. International Journal of Advanced Research (2015), Volume 3, Issue 12, 812 – 821. http://www.journalijar.com

[2] Sherine Maher Rizk  and  Nagwa Ali Sabri. Evaluation of clinical activity and safety of Daflon 500 mg in type 2 diabetic female patients. Saudi Pharmaceutical Journal (2009) 17, 199–207.  doi:10.1016/j.jsps.2009.08.008. 

[3] Sara AR, Mohamed Raslan, Eslam M Shehata and Nagwa A Sabri. Impact of Applied Protective Measures of COVID-19 on Public Health. Acta Scientific Pharmaceutical Sciences 5.7 (2021):63-72.

[4] Marwa M Nagib, Mariane G Tadros, Hadwa Ali Abd Al-khalek, Rania M Rahmo, Nagwa Ali Sabri, Amani E Khalifa, Somaia I Masoud. Molecular mechanisms of neuroprotective effect of adjuvant therapy with phenytoin in pentylenetetrazole-induced seizures: Impact on Sirt1/NRF2 signaling pathways. Neurotoxicology Vol. 68, 47-65, 2018. https://doi.org/10.1016/j.neuro.2018.07.006.

[5] Amr A. Elsayed , Esther T. Menze, Mariane G. Tadros, Bassant M. M. Ibrahim,
Nagwa A. Sabri , Amani E. Khalifa.
Naunyn-Schmiedeberg's Arch Pharmacol (2018). Effects of genistein on pentylenetetrazole-induced behavioral and neurochemical deficits in ovariectomized rats 391:27–36 https://doi.org/10.1007/s00210-017-1435-7.

7- An information concerning prevalence and incidence all over the world was missed , the authors mentioned two countries only , better to give an overview allover the world as possible.

Author Response

Reply to Reviewer #2

IMPORTANT:  Manuscript with corrections also of English by the MDPI linguistic service

1-The abstract better to be divided into background, aim, method, results, and conclusion.
2-The following sections were absent; methods and Results although they are present as information but not under methods and results.
ANSWER
Thank you for your suggestion. I divided the abstract into the various sections you recommended.

3-Captions of tables and figures required to be written in full details explaining the target from the table or figure, the importance and effect of information on the PD.
ANSWER
Thank you for your suggestion.
I re-formulated the title of the tables and of the figure, making their meanings and objectives more explicit.

4- Effect of age and psychiatric disorders on the progression of PD better to be mentioned in the review.
ANSWER
Thank you for your suggestion.
I have added text to the manuscript concerning the two conditions you indicated.

5-Study flow chart for the review is missed which would be much better to be designed and submitted with the review as a summarized explanatory chart for the methodology.
ANSWER
Thank you for your suggestion. I have inserted an “Explanatory Flow Chart of the revision methodology” in the manuscript.

6-The insulin resistance and glycemic control are important etiology of many inflammatory diseases. Hereunder paragraphs containing  some antioxidants with their mechanisms  as silymarin, diosmin,hesperidin, coenzyme Q10 and alpha-tocopherol, and genistein. So, the following paragraph to be added to the introduction of the manuscript ;

"Suboptimal glycemic control and insulin resistance are associated with high risk of
macrovascular and microvascular complications. Silymarin is a herbal medicine with an antioxidant and anti-inflammatory properties when given in a dose of 140mg thrice daily for 3 months as an adjuvant for glycemic control, lipid profile and insulin resistance proved to have a beneficial efficacy. [1] Additionally, Daflon 500 mg (micronized purified flavonoid fraction of Rutaceae aurantiae, consisting of 90% diosmin and 10% hesperidin), twice daily for 45 days is helpful in reducing glucose level and the risk of cardiovascular disease. [2]

Additionally,  herbs, and healthy diet including fruits and vegetables, can supply the body with beneficial nutrients and antioxidants [3] including coenzyme Q10 and alpha-tocopherol which have a neuroprotective as they proved to have protective effects of antioxidants [4]. Moreover, estrogenic compounds as genistein proved to exhibit a neuroprotective effect attributed to its estrogenic, antioxidant, and/or anti-apoptotic properties [5].

ANSWER
Thank you for your suggestion.
The articles you have reported to me are very interesting. For reasons of space in the manuscript and for its large list of references, I had to limit the part of the text that I added and I inserted 2 new references, which you have indicated in the bibliography.
In any case, I have included in the text part of the text that you have indicated to me and which I believe is more strictly relevant to the specific topic of my manuscript.

7- An information concerning prevalence and incidence all over the world was missed , the authors mentioned two countries only , better to give an overview allover the world as possible.
ANSWER
Thank you for your suggestion. I added new information on the prevalence of PD in the text.

------------------------------------------------------------------------------